# Frailty Levels In Geriatric Hospital paTients (FLIGHT)—the prevalence of frailty among geriatric populations within hospital ward settings: a systematic review protocol

Paul Doody,[1] Justin Aunger,[1] Evans Asamane,[1] Carolyn A Greig,[1,2] Janet Lord,[3] Anna Whittaker[1,4]

[1]School of Sport, Exercise and Rehabilitation Sciences, University of Birmingham, Birmingham, UK
[2]MRC—Arthritis Research UK Centre for Musculoskeletal Ageing Research, Birmingham, UK
[3]NIHR Surgical Reconstruction and Microbiology Research Centre, University of Birmingham, Birmingham, UK
[4]Faculty of Health Sciences and Sport, University of Stirling, Stirling, UK

**Correspondence to**
Paul Doody;
p.d.doody@bham.ac.uk

## ABSTRACT

**Introduction** Frailty is a common and clinically significant condition in geriatric populations, associated with adverse health outcomes such as hospitalisation, disability and mortality. Although there are systematic reviews/meta-analyses assessing the prevalence of frailty in community-dwelling older adults, nursing home residents, and cancer and general surgery patients, there are none assessing the overall prevalence of frailty in geriatric hospital inpatients.

**Methods and analysis** This review will systematically search and analyse the prevalence of frailty within geriatric hospital inpatients within the literature. A search will be employed on the platforms of Ovid, Web of Science and databases of Cumulative Index to Nursing and Allied Health Literature (CINAHL) Plus, SCOPUS and the Cochrane Library. Any observational or experimental study design which utilises a validated operational definition of frailty, reports the prevalence of frailty, has a minimum age ≥65 years, attempts to assess the whole ward/clinical population and occurs in hospital inpatients, will be included. Title and abstract and full-text screenings will be conducted by three reviewers. Methodological quality of eligible studies will be assessed using the Joanna Briggs Institute critical appraisal tool. Data extraction will be performed by two reviewers. If sufficient data are available, a meta-analysis synthesising pooled estimates of the prevalence of frailty and pre-frailty, as well as the prevalence of frailty stratified by age, sex, operational frailty definition, prevalent morbidities, ward type and location, among older hospitalised inpatients will be conducted. Clinical heterogeneity will be assessed by two reviewers. Statistical heterogeneity will be assessed through a Cochran Q test, and an $I^2$ test performed to assess its magnitude.

**Ethics and dissemination** Ethical approval was not required as primary data will not be collected. Findings will be disseminated through publication in peer reviewed open access scientific journals, public engagement events, conference presentations and social media.

**PROSPERO registration number** 79202.

## Strengths and limitations of this study

► First review to systematically or exclusively assess the overall prevalence of frailty in geriatric hospital inpatients.
► Will seek to provide stratified analysis of the prevalence of frailty based on age, sex, operational frailty definition, prevalent morbidities, ward type and location.
► Three independent reviewers during screening phase; ensuring high internal reliability and consistency of included studies.
► Will include only studies for which the full text is available in English, therefore will likely be relatively over-representative of Western nations (Europe, Australasia and the Americas); although this is true of scientific publications in general.

## INTRODUCTION

Frailty is a common and clinically significant condition within geriatric populations,[1] predominantly due to its association with adverse health outcomes, such as hospitalisation, disability and mortality.[1–6] Although there are systematic reviews and meta-analysis assessing the prevalence of frailty among community-dwelling older adults,[7–10] nursing home residents,[11] and cancer[12] and general surgery patients,[13] presently there are no systematic reviews or meta-analysis which assess the overall prevalence of frailty among geriatric hospital inpatients. This constitutes an important gap in the literature which needs to be addressed and has important consequences. Such consequences include the tailoring of services within this setting to the needs of service users, for example, the potential implementation of exercise rehabilitation treatments within this setting for this cohort; with physical activity and exercise

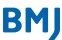

being proposed as potentially offering the best form of treatment for frail older adults,[14] and shown to be capable of reducing and even reversing frailty within older adults.[15 16] Through providing a highly detailed analysis of the prevalence of frailty among older populations within this setting, this review has the potential to aid in the facilitation of improvements in the planning and orientation of organisational structures and resources, to meet the needs of this population and enhance the care of frail older adults in inpatient hospital settings.

## METHODS AND DESIGN
### Review aim
The aim of this review is to systematically search and analyse the prevalence of frailty among geriatric populations (aged ≥65 years) within inpatient hospital settings within the literature. If a meta-analysis proves possible, the aim of this study is also to synthesise pooled estimates of the prevalence of frailty and pre-frailty, as well as the prevalence of frailty stratified by age, sex, operational frailty definition, prevalent morbidities, ward type and location (country and continent), among hospital inpatients.

### Review objectives
1. To identify and compare studies reporting the prevalence of frailty within hospital ward settings.
2. To combine the extracted data to calculate the pooled overall prevalence of frailty in hospitalised geriatric inpatients.
3. To perform stratified analysis of the prevalence of frailty based on age, sex, operational frailty definition, prevalent morbidity and ward type in order to assess the relationship between frailty and these factors.

### Eligibility criteria
Inclusion criteria: all studies must have a minimum age of ≥65 years, use a clearly defined and validated operational definition for the classification of frailty (ie, one which takes into consideration the multi-dimensional nature of the condition, and has been specifically validated for the assessment of frailty; either through comparison with existing validated tools or its predictive value regarding negative health outcomes aligned with frailty), either assess (or attempt to assess) the whole ward, department, unit, hospital or specific clinical population, or employ some form of randomised selection of participants, occur within a hospital setting, in, or including, hospital inpatients (operationally defined as any patient admitted to hospital who remains overnight, or were initially expected to remain overnight), report the prevalence of frailty or provide sufficient data to allow the calculation of the prevalence of frailty. If a study examines a mixed cohort, only data relating to hospital inpatients will be included in the review.

Exclusion criteria: all studies not written in English, studies where the sample are not hospital inpatients (ie, outpatients, day patients or community-dwelling individuals).

### Information sources
Searches will be conducted on the platforms of Ovid (incorporating the databases of Journals@Ovid full text, EMBASE, CAB abstracts, Ovid MEDLINE In process and other non-indexed citations, Ovid MEDLINE, and PyschINFO) and Web of Science (incorporating the databases of Science Citation Index Expanded (SCI-Expanded), Conference Proceedings Citation Index—Science (CRI-S), and Emerging Sources Citation Index (ESCI)), and the databases of Cumulative Index to Nursing and Allied Health Literature (CINAHL) Plus, SCOPUS and the Cochrane Library databases (the Cochrane Database of Systematic Reviews (CDSR), the Cochrane Central Register of Controlled Trials (CENTRAL), the Cochrane Methodology Register (CMR), the Database of Abstracts of Reviews of Effect (DARE), Health Technology Assessment database (HTA), and the NHS Economic Evaluation Database (EED)).

### Types of studies
Any form of observational or experimental study design which assesses the prevalence of frailty and meets the above eligibility criteria. For longitudinal observational studies, and experimental studies, frailty scores and additional data will be extracted from baseline data, provided baseline data meets the above eligibility criteria.

### Search strategy
The search strategy will be conducted on the two platforms of Ovid and Web of Science, as well as the databases of SCOPUS, CINAHL Plus and the Cochrane Library databases (online supplementary appendix 1). These searches will encompass all available literature published prior to 21 November 2018.

### Screening
Prior to the commencement of title and abstract screening by the three independent reviewers, duplicates will be removed using EndNote V.X8.2. The reduced list of studies will be manually screened for the removal of any remaining duplicates. All reviewers will be provided with an instructional screening form (online supplementary appendix 2), and a .ris file containing all studies captured within database searches. The screening form will list the eligibility criteria and instructions on setting up the .ris file for screening within a reference manager.

The title and abstract of all studies will then be independently screened by the three reviewers, with each reviewer placing potentially eligible studies into a separate folder. On completion, potentially eligible studies from all three reviewers will be placed into a 'master folder' and the results collated. Duplicates will be removed, leaving the final combined list of studies for the full text screening phase. All reviewers will then independently screen the full text of remaining studies utilising the screening form and maintain separate files for included

and excluded studies (including reasons), as well as for studies for which the reviewer feels the need to contact the authors for clarification or additional information.

On completion, a full text screening master file (online supplementary appendix 3) will be formulated by the lead reviewer displaying each reviewer's full text screening decision for each study. All three reviewers will then meet to discuss the decisions of each study and endeavour to come to an agreement on studies for which there is not initial unanimous consensus. During this process a full list of included and excluded studies (with reasons), and studies for which reviewers agree to contact authors for additional information or clarification will be formed by the lead reviewer. The lead reviewer will then contact study authors and, on receipt of clarification or additional information, will meet with reviewers to discuss the inclusion/exclusion of the study.

Manual screening will also be employed by reviewers and include the reference lists of all included studies, as well as excluded but potentially relevant studies or systematic reviews captured within the screening. As part of the grey literature search of this review, in process publications will also be searched and conference abstracts will be followed up with authors to ascertain if a full text relating to the data is available. Studies of the same cohort will be included only once, using the study which provides the most information about the cohort relevant to this review.

### Assessment of methodological quality

The quality of eligible studies from full text screening will be assessed by two reviewers independently using the Joanna Briggs Institute critical appraisal tool for studies reporting prevalence data[17]. In the event of any discrepancies between the two reviewers, a consensus will be attempted to be reached by discussion. In the event, a full consensus cannot be reached between the two reviewers after an exhaustive discussion, the opinion of a third reviewer will be obtained, and the proceeding majority consensus will be taken.

### Data extraction

Data extraction will be performed by two reviewers independently. In the event of any discrepancies between the two reviewers, a consensus will be attempted to be reached by discussion. In the event that a full consensus cannot be reached between the two reviewers after an exhaustive discussion, the opinion of a third reviewer will be obtained and the proceeding majority consensus will be taken.

The following data, where available, will be extracted from all eligible studies (see online supplementary appendix 4 for template). If any data are not immediately available, the authors of the studies in question will be contacted in an attempt to retrieve all applicable data:

Study details: authors, year of publication, study title, journal of publication and aim. Study methods: setting, ward/department/unit/hospital type/clinical population, study design, recruitment duration, subject

characteristics (age of participants (mean and SD, range)), sex (proportion of male/female participants), country/continent, sample size, diagnosis/prevalent morbidity (if applicable), any other relevant characteristics), criteria utilised for the operational definition of frailty. Results: Number of frail participants, number of 'pre-frail' participants, number of robust/non-frail participants, prevalence of frailty, prevalence of pre-frailty, prevalence of robustness/non-frailty, number of male participants, number of frail male participants, number of pre-frail male participants, number of non-frail/robust male participants, prevalence of frailty in male participants, prevalence of pre-frailty in male participants, prevalence of non-frailty/robustness in male participants, number of female participants, number of frail female participants, number of pre-frail female participants, number of non-frail/robust female participants, prevalence of frailty in female participants, prevalence of pre-frailty in female participants, prevalence of non-frailty/robustness in female participants, and finally authors comments and reviewers' comments.

External to the studies, data will also be extracted with regard to the 5-year average gross domestic product (GDP) per capita purchasing power parity (PPP) (current international $) of the country in which each study takes place, incorporating the 5 years directly preceding the commencement of recruitment to the study.[18] External data will also be extracted with regard to the 5-year average healthcare expenditure per capita PPP (current international $) of the country in which each study takes place, incorporating the 5-years directly preceding the commencement of recruitment to the study.[19] Each calendar year of the study will also be included provided recruitment continues through to >6 months in the preceding year.

### Data synthesis

Quantitative synthesis (meta-analysis): if a sufficient quantity of identified studies are comparable, a meta-analysis, pooling the aggregated data from each study, will be performed. Clinical heterogeneity will be assessed by two reviewers based on their judgement of the available data and any disagreements will be discussed thoroughly with the aim of reaching a unanimous consensus. If a unanimous consensus cannot be reached, the opinion of a third reviewer will be sought, and the proceeding majority consensus will be taken. Statistical heterogeneity will be assessed through the utilisation of a Cochran Q test and considered present at $p < 0.05$. An $I^2$ test will be performed in order to assess the magnitude of this heterogeneity, with $I^2$ values of 25%, 50% and 75% being considered low, moderate and high, respectively. If the Cochrane Q statistic test detected statistically significant heterogeneity, combined with the researcher's assessment, a randomised-effects model will be used. Given the nature of this review and in particular its overall aim, combined with the eligible

studies identified in preliminary searches, it is likely the initial quantitative synthesis will use a random-effects model.

Stratified analysis will also be conducted according to age (65–74 years, 75–84 years and 85+ years), sex, operational frailty definition, ward type, prevalent morbidity and location (country and continent) where possible. These variables have been specifically chosen for stratified analysis predominantly due to an enhanced knowledge of these areas being of practical utility to researchers and clinicians; stemming from empirical evidence persistently showing alterations in these factors to impact on the prevalence of frailty.[2 4 20–22] As such stratified analysis pertaining to these variables will facilitate this review to provide a more in-depth and thorough insight into the prevalence of frailty among geriatric hospital inpatients.

Clinical heterogeneity for stratified analysis will be assessed by two reviewers based on their judgement of the available data. Any disagreements will be discussed thoroughly with the aim of reaching a unanimous consensus. If a unanimous consensus cannot be reached, the opinion of a third reviewer will be sought. Statistical heterogeneity for sub-analysis will similarly be assessed through the utilisation of a Cochran Q test and considered present at $p<0.05$. An $I^2$ test will be performed in order to assess the magnitude of this heterogeneity, with $I^2$ values of 25%, 50% and 75% being considered low, moderate and high, respectively.

Similarly, it is likely a random-effects model will be utilised to synthesise pooled estimates of the prevalence of frailty stratified by these criteria (although there is more of a likelihood that a fixed effects model could potentially be utilised within these analyses, in comparison to the initial analysis, given the nature of stratified analysis).

Correlation analysis will also be employed to examine the relationship between the prevalence of frailty of geriatric inpatients and economic prosperity (GDP per capita PPP) (current international $), and healthcare expenditure (per capita PPP) (current international $). In addition, multi-linear regression analysis will examine the predictive value between economic prosperity and healthcare expenditure and the prevalence of frailty of geriatric inpatients. Preliminary research into these areas have shown frailty in the community to be correlated with economic indicators (GDP per capita PPP),[23] however, note that more research is needed in this regard to better understand this relationship; which this review will facilitate through examination of the relationship of GDP per capita PPP and healthcare expenditure, and the prevalence of frailty among geriatric hospital inpatients.

Qualitative synthesis: if a meta-analysis is not possible based on the nature of the studies and the data available, a more thorough systematic narrative analysis will be conducted, with findings presented in both textual and tabular formats.

## Patient and public involvement

All authors are strong proponents of patient and public involvement and engagement with research and believe the finding of this review will be important to aid the facilitation of improvements in the planning and orientation of organisation structures and resources within this setting to meet the needs of service users; specifically relating to the enhanced care of older adults in inpatient hospital settings. However, given the nature of this study (systematic review), it was not possible to involve the public. However, the findings will be disseminated to our patient and public involvement groups.

## Ethics and dissemination

Formal ethical approval was not required for this review as primary data will not be collected. The findings of this study will be disseminated through publication in the form of scientific papers in peer reviewed open access scientific journals, public engagement events within the UK and Europe, online via social media (Twitter, Instagram) and the PANINI project website,[24 25] and presentation at conferences within the UK and internationally. This review is scheduled for completion during the second half of 2019.

**Acknowledgements** The authors of this review would like to thank and acknowledge Ms. Lynne Harris (Subject advisor of the Main Library at the University of Birmingham, United Kingdom) for her assistance during the formulation of the search strategy utilised within this systematic review.

**Contributors** PD is guarantor and lead reviewer. PD designed the systematic review protocol, conducted the literature searches and prepared this manuscript for publication, with supervision, input and feedback from AW, CAG and JL. EA and JA are independent reviewers for title and abstract and full text screenings. JA will also act as independent data extractor for included studies. All authors have read and approved the final manuscript.

**Funding** This review has been funded by the European Commission's Horizon 2020 research and innovation programme under the Marie Sklodowska-Curie grant agreement (675003); of which PD, EA and JA are Marie Sklodowska-Curie Doctoral Research Fellows, AW, JL and CAG doctoral supervisors, and AW the grants Principal Investigator.

**Competing interests** None declared.

**Patient consent for publication** Not required.

**Provenance and peer review** Not commissioned; externally peer reviewed.

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
