## [Reviewer comments · BMJ Open]

ARTICLE DETAILS

TITLE (PROVISIONAL)	Frailty Levels in Geriatric Hospital paTients (FLIGHT) - The prevalence of frailty amongst geriatric populations within hospital ward settings: A systematic review protocol
AUTHORS	Doody, Paul; Aunger, Justin; Asamane, Evans; Greig, Carolyn; Lord, Janet; Whittaker, Anna

VERSION 1 – REVIEW

REVIEWER	Han Ting Wang Universite de Montreal, Montreal, Quebec, Canada
REVIEW RETURNED	10-Apr-2019

GENERAL COMMENTS	In this protocol, Dr Doody et al. wants to adress an important gap in the knowledge of frailty. the overall protocol is adequate to answer the question. here are some specific comments for the authors to consider: study population: we are looking at all studies where they assessed frailty in-hospital. the prevalence of frailty varies by underlying pathology. considering this, some specific populations need to be taking into consideration: 1 - intensive care units patients: they are among the sickest of all patients. there have already been previous systematic reviews on this subject. also, ICU patients are from different ward and go to different ward. i would have suggested to exclude this subgroup. 2 - surgical patients. most surgical papers would have assessed frailty preoperatively and not during hospitalization. will they be included? having being performing a systematic review myself on frailty (not in this setting), there have been an increase in number of studies. what i fear is the feasibility of this study. there might be a lot of titles and abstracts! maybe the preferred way would be to focus on medical wards? which still includes an important variety of patients. types of studies. authors mention including observational and experimental studies. one consideration are studies targeting the frail patients per se. some trials included only frail patients or a subset of these patients for intervention. this will give a biased prevalence of frailty on the ward. a point the authors should consider and maybe mention on how they would adress this. this is also true for observational studies where they might be interested only in the pre-frail or frail
--

	patients. i would suggest to exclude them since we are interested in the prevalence of frailty. search methods: will there be different search methods and keywords for observational and experimental studies? what will the authors do with conference abstracts for example. will they be included if enough data can be extracted or will they all be excluded? please specify substudies it is possible that an investigator, using one database publishes different papers with varying objective but always the same cohort. will only the initial study be included? or all of them? finally, this is a major undertaking. are the authors interested into assessing the association of frailty with some form of outcomes? or look at interventions? considering the amount of work put into the screening and going through all the titles, and abstracts, it could be interesting to add this aspect. i hope my comments will be helpful to the authors. thank you!
--	--

REVIEWER	Katarina Wilhelmson Institute of Neuroscience and Physiology, Sahlgrenska Academy, and Centre of Aging and Health-AgeCap, University of Gothenburg, Gothenburg, Sweden
REVIEW RETURNED	11-Apr-2019

GENERAL COMMENTS	This study protocol describes a systematic review with the aim to analyse the prevalence of frailty amongst geriatric populations within inpatient hospital settings. If meta-analyses proves possible, they will also estimate the prevalence of pre-frailty, and frailty stratified by age, sex, frailty definition, morbidity, ward type and location. I find the topic interesting and the knowledge from the review will be important. There is need of some clarifications: How is “geriatric population” defined? Only by age 65 years and over? Are patients 65 years and older the same as geriatric patient? What do you mean by “geriatric hospital inpatients”? Is it only patients admitted to a geriatric ward, or is it all patients age 65 years and over admitted to a hospital? First, I interpreted it as a patient in a geriatric ward, but later on I assume you mean a “geriatric patient”= 65 years and older, admitted to any hospital ward. This needs to be more clearly described. If it is any hospital ward, are for example surgical wards included? What do you mean by “location”? When will the study be done? How long time is it planned to take before it is completed? Page 6. Search strategy: for which years will the search be done? Page 8. Data synthesis: “If a sufficient quantity of identified studies are comparable....” How will you determine if this is accomplished? How many studies are needed, and how will you decide if they are comparable?
--

	Page 9, line 181-182: why only two age-groups? I would prefer at least three, 65-74, 75-84 and 85+. The prevalence of frailty might/is very likely to be higher for 85+ than 75+. Which types of wards and morbidity will be included? The limitations of the study is stated, but not discussed (how will the limitations affect the results and the interpretation of the results of the study?).
--	---

REVIEWER	Dr Rebecca L. Gould Senior Research Fellow, Division of Psychiatry, University College London, London, UK
REVIEW RETURNED	11-Apr-2019

GENERAL COMMENTS	This is a well written protocol for a systematic review in an important area. My comments are shown below: Strengths and limitations: 1) It says 3 independent reviewers will be involved, but 3 reviewers will only be involved in screening titles and abstracts. The most important tasks in a systematic review are data extraction and quality appraisal, and only 2 reviewers will be involved in these processes. So it is a misrepresentation to say that a strength of the study is “3 reviewers, ensuring high internal reliability and consistency”. This point needs to be removed as having 2 reviewers for the data extraction and quality appraisal tasks (with a third independent person to discuss any discrepancies that cannot be resolved through discussion with the 2 reviewers) is standard practice for systematic reviews (and the minimum requirement according to PRISMA guidelines). It is not a particular strength of this study. Instead, the authors could mention the comprehensive search of a wide range of literature sources being a strength of the study. 2) The last point is no different from the first point and therefore should be removed. 3) The limitation of only including studies in English should be expanded upon – what will this mean for the interpretation of results and any conclusions drawn? Eligibility criteria: 1) The authors note that all studies must have a minimum age of ≥ 65 years. Imagine that a study does not report a minimum age range, but instead reports a mean age of 80 years, will the study be excluded? Studies frequently report mean ages rather than age range, and they may not report age as an inclusion criterion if they've examined geriatric inpatient wards (e.g. they might just include anybody on the geriatric inpatient wards, and wards may vary in terms of how they define “geriatric” e.g. some may say 60 years and over). The authors may want to consider specifying a minimum mean age, as well as the minimum age, or at the very least acknowledge the limitation of this approach. 2) How are the authors defining “a clearly defined and validated operational definition for the classification of frailty”? Do they mean use of a validated frailty assessment tool, a self-rating questionnaire, etc? This needs to be discussed in more detail. 3) The authors give outpatients, day patients or community-dwelling individuals as examples of non-hospital inpatients. Presumably they mean non-hospital outpatients? Information sources:
---

	1) The authors need to provide the years that will be searched for each of the databases. 2) There is no mention of manual searching (e.g. hand searching of references of relevant studies and literature reviews). Why are the authors not doing this? This is standard practice as some older studies may not have been entered onto the databases. Data extraction: 1) Why are the authors not extracting data regarding ethnicity? Some studies may not provide these data, but this doesn't mean that you should not attempt to extract these data. 2) Presumably the authors will extract data regarding the criteria utilised for the operational definition of pre-frailty, as well as frailty, given that they are also interested in the pre-frail state? Data synthesis: 1) Why did the authors choose age, sex, frailty definition, ward type, prevalent morbidity and location as factors for the stratified analysis? Are these evidence-based factors that have been shown to influence prevalence rates in other frailty populations? There needs to be more justification for this choice of variables, either in the Introduction or Methods section. 2) More information needs to be provided about how some of the variables will be stratified. Will frailty definition be stratified by frailty model (e.g. cumulative deficits model vs. Fried model) or assessment mode (e.g. self-rating vs. clinician-rating of frailty)? How will prevalent morbidity be stratified? Does location mean country? How will this be stratified? 3) More information needs to be provided about how they will complete the stratified analysis (e.g. subgroup analysis, metaregression, etc).
--	---

VERSION 1 – AUTHOR RESPONSE

Reviewer: 1

Reviewer Name: Han Ting Wang

Institution and Country: Universite de Montreal, Montreal, Quebec, Canada

Comment - Dear BMJ editor, thank you for letting me review this protocol manuscript. In this protocol, Dr Doody et al. wants to adress an important gap in the knowledge of frailty. the overall protocol is adequate to answer the question.

Response - Thank you for your review of this manuscript, and positive comments.

Comment: here are some specific comments for the authors to consider:

study population:

we are looking at all studies where they assessed frailty in-hospital.

the prevalence of frailty varies by underlying pathology. considering this, some specific populations need to be taking into consideration:

1 - intensive care units patients: they are among the sickest of all patients. there have already been previous systematic reviews on this subject. also, ICU patients are from different ward and go to different ward. i would have suggested to exclude this subgroup.

2 - surgical patients. most surgical papers would have assessed frailty preoperatively and not during hospitalization. will they be included? If they are assessed for frailty in ICU we would consider them an ICU patient irrespective or trnafert to or from a previous or subsequent ward.

Response - We agree with the Reviewer that prevalence of frailty will vary dependent on the specific sub-type of hospital inpatient. While providing a combine pooled overall estimate of the prevalence of frailty, this review will additionally provide estimates relating to specific morbidities and ward type; to provide a more comprehensive insight into the overall prevalence of frailty of all geriatric hospital inpatients. For the purpose of this review the operation definition of “inpatient” has been defined as “any patient admitted to hospital and whom either remained overnight, or were initially expected to remain overnight”. This has now been added to the manuscript on Lines 96 - 97 (Additive text in bold): “occur within a hospital setting, in, or including, hospital in-patients (operationally defined as any patient admitted to hospital and whom either remains overnight, or were initially expected to remain overnight),”

Any patient which falls within this definition will be included, including ICU and surgical patients, as while they are specific clinical populations, they are hospital inpatients.

Presumably the systematic review to which the Reviewer refers is Muscedere et al. 2017, which although occurring in ICU patients, had a minimum age of 18 years and found that 30% of ICU patients were frail. Geriatric inpatient are typically more frail than this, and this review did not report the prevalence of frailty for patients over the age of 65 years, or geriatric patients in general, and of the 10 studies within this review, only 3 are eligible for this present review, however more have been included. This present review will provide a more detailed description of the prevalence of frailty specifically in geriatric hospital inpatients, both within the ICU during stratified analysis, and overall within this setting.

In regard to surgical patients, only studies where the patients have been assessed while they were inpatients i.e. admitted with inpatient status will be included, as per the eligibility requirements on Line 95 - 96 of the manuscript which reads “occur within a hospital setting, in, or including, hospital in-patients”, which hopefully has been further clarified by the inclusion of the review’s operational definition of inpatient from Lines 96 – 97 to subsequently read “(operationally defined as any patient admitted to hospital who remains overnight, or who was initially expected to remain overnight”. As such surgical patients or any scenario where patients are assessed for frailty as outpatients, day-patients, prior to admission, or if a specific frailty screening tool was implemented as inpatients but designed to assess pre-admission frailty status, will be excluded during the screening process.

Patients screened for frailty in the ICU will be considered ICU inpatients in this review irrespective of transfer to subsequent wards if this is when and where the screening took place.

Therefore, we would respectfully disagree with the reviewer’s suggestion to exclude these inpatients. All geriatric hospital inpatients will be included in the main analysis of this review as they are in fact members of this population. However, the exact prevalence will be further investigated with specific stratified analysis based on specific characteristics including ward, department, unit, hospital type or clinical population.

Comment - having being performing a systematic review myself on frailty (not in this setting), there have been an increase in number of studies. what i fear is the feasibility of this study. there might be a lot of titles and abstracts! maybe the preferred way would be to focus on medical wards? which still includes an important variety of patients.

Response - The authors have now successfully screened 3004 title and abstracts and 664 full text studies and are presently emailing authors for clarification on possible inclusion of studies and completing the quality assessment of previously identified eligible studies; which the reviewer is correct, has been a significant undertaking by the three reviewers. This study will include stratified analysis of all types of medical wards present within the review.

Comment - types of studies.

authors mention including observational and experimental studies. one consideration are studies targeting the frail patients per se. some trials included only frail patients or a subset of these patients for intervention. this will give a biased prevalence of frailty on the ward. a point the authors should

consider and maybe mention on how they would address this. This is also true for observational studies where they might be interested only in the pre-frail or frail patients. I would suggest to exclude them since we are interested in the prevalence of frailty.

Response – We agree with the Reviewer and have already included this in the eligibility criteria of this review, located on Lines 90 – 101. These studies mentioned by the reviewer i.e. “trials only including frail patients or a subset of these patients for intervention” will be excluded based on the eligibility criterion or “assess or attempt to assess the whole ward or clinical population or employ some form of randomised selection of participants”. Studies which include only frail patients, will be included if they can provide evidence to satisfy the above criteria, for example in the event that initially all patients were screened for inclusion based on frailty status, and these data are available.

Comment - search methods:

Will there be different search methods and keywords for observational and experimental studies?

Response - The search strategy employed on the data bases is a single search strategy for the review, the description of which is located on lines 120 – 123, and the exact implemented search strategies within Appendix 1. The authors are confident that this will capture all study types and has been successful in identifying all preliminary studies which were identified prior to systematic searches being implemented for both observational and experimental study designs alike.

Comment - what will the authors do with conference abstracts for example. Will they be included if enough data can be extracted or will they all be excluded? Please specify

Response - In the case of conference abstracts, the authors of these studies will be contacted for a full text. If an English full text is not available, they will be excluded as per the requirements of full text screening. This has been incorporated into the manuscript on lines 148 – 150:

“As part of the grey literature search of this review, in process publications will also be searched and conference abstracts will be followed up with authors to ascertain if a full text relating to the data is available”.

Comment - substudies

It is possible that an investigator, using one database publishes different papers with varying objectives but always the same cohort. Will only the initial study be included? or all of them?

Response - Studies of the same cohort will be included only once. The study which provides the most information about the cohort relevant to this review will be included and the others excluded with the justification that the cohort has already been included in the review. Authors will be contacted if there is any degree of ambiguity as to whether two studies have included the same sample of participants. This has been clarified in the manuscript on line 150 – 152:

“Studies of the same cohort will be included only once, using the study which provides the most information about the cohort relevant to this review.”

Comment - finally, this is a major undertaking. Are the authors interested in assessing the association of frailty with some form of outcomes? or look at interventions?

Response - This review will relate solely to the prevalence of frailty within geriatric hospital inpatients – to associate this with the efficacy of an intervention, we feel, would require a separate review. However the authors are interested in, and have now included a component within the review, assessing the association between the economic prosperity, and health care expenditure of countries, and the prevalence of frailty of geriatric hospital inpatients, which has been included from lines 183 – 189 and lines 222 – 226 of the manuscript (additive text outlined below) and within the data extraction form in Appendix 5 (Columns P and Q of spreadsheet one, and spreadsheet 2 and 3 in their entirety). Lines 183 – 189:

“External to the studies, data will also be extracted with regard to the 5 year average Gross Domestic Product (GDP) per capita Purchasing Power Parity (PPP) (current international \$) of the country in

which each study takes place; incorporating the five year directly preceding the commencement of recruitment to the study (International Monetary Fund, 2019). External data will also be extracted with regard to the 5 year average health care expenditure per capita PPP (current international \$) of the country in which each study takes place, incorporating the five years directly preceding the commencement of recruitment to the study (World Health Organisation, 2019). Each calendar year of the study will also be included provided recruitment continues through to > 6 months in the preceding year.”

Lines 222 - 226:

“Correlation analysis will also be employed to examine the relationship between the prevalence of frailty of geriatric inpatients and economic prosperity (GDP per capita PPP - current international \$), and health care expenditure (per capita PPP - current international \$). Additionally, multi-linear regression analysis will examine the predictive value between the economic prosperity and health care expenditure, and the prevalence of frailty amongst geriatric hospital inpatients.”

References:

International Monetary Fund, (. 2019, April-last update, World Economic Outlook (April 2019).

Available: <https://www.imf.org/external/datamapper/PPPPC@WEO/ADVEC/WEOWORLD> [2019, 02/05/].

World Health Organisation, (. 2019, April-last update, Global Health Expenditure Database (GHED).

Available: <http://apps.who.int/nha/database/Home/Index/en> [2019, 02/05/].

i hope my comments will be helpful to the authors.

thank you!

Response - Thank you for your comments. The authors hope the above responses adequately address these and thank you again for your review.

Reviewer: 2

Reviewer Name: Katarina Wilhelmson

Institution and Country: Institute of Neuroscience and Physiology, Sahlgrenska Academy, and Centre of Aging and Health-AgeCap, University of Gothenburg, Gothenburg, Sweden

Comment - This study protocol describes a systematic review with the aim to analyse the prevalence of frailty amongst geriatric populations within inpatient hospital settings. If meta-analyses proves possible, they will also estimate the prevalence of pre-frailty, and frailty stratified by age, sex, frailty definition, morbidity, ward type and location. I find the topic interesting and the knowledge from the review will be important.

Response - Thank you for your review of this manuscript.

There is need of some clarifications:

How is “geriatric population” defined? Only by age 65 years and over? Are patients 65 years and older the same as geriatric patient?

Response - Within this review the term geriatric is operationally defined as aged 65 years or over, as alluded to within the studies inclusion criteria on Line 91: “a minimum age \geq 65 years”; as well as within the methods and analysis section of the abstract on Line 31: “has a minimum age \geq 65 years”. To further clarify we have now specified this upon the first in text reference of the term “geriatric population” within the methods section on line 78 (additive text in bold):

“The aim of this review is to systematically search and analyse the prevalence of frailty amongst geriatric populations (aged \geq 65 years) within inpatient hospital settings within the literature.”

Comment - What do you mean by “geriatric hospital inpatients”? Is it only patients admitted to a geriatric ward, or is it all patients age 65 years and over admitted to a hospital? First, I interpreted it as a patient in a geriatric ward, but later on I assume you mean a “geriatric patient”= 65 years and older, admitted to any hospital ward. This needs to be more clearly described. If it is any hospital ward, are for example surgical wards included?

Response – The term geriatric hospital inpatient refer to any inpatients aged 65 years or older. The authors hope that the previous statements help clarify this point. Provided the eligibility criteria are met (Lines 90-101), surgical wards will also be included (provided patients are screened for frailty as inpatients and not prior to admission as per the aforementioned eligibility criteria), as will all hospital wards which included geriatric inpatients and meet the eligibility criteria of the review to give a truly representative estimate of the prevalence of frailty amongst geriatric hospital inpatients as well as a thorough insight into this through stratified analysis.

Comment - What do you mean by “location”?

Response - Location specifically refers to the country and continent on which the study takes place. As such the prevalence of frailty will be stratified by country and continent, in a manner consisted with the data extraction form attached to this application (Appendix 5 – column N and O respectively). We have also now changed “country/location” to “country/continent” on line 173.

The authors have now also included a component within the review, assessing the association between the economic prosperity, and health care expenditure of countries, and the prevalence of frailty of geriatric hospital inpatients, which has been included from lines 183 – 189 and lines 222 – 226 of the manuscript (additive text outlined below) and within the data extraction form in Appendix 5 (Columns P and Q of spreadsheet one, and spreadsheet 2 and 3 in their entirety).

Lines 183 – 189:

“External to the studies, data will also be extracted with regard to the 5 year average Gross Domestic Product (GDP) per capita Purchasing Power Parity (PPP) (current international \$) of the country in which each study takes place; incorporating the five year directly preceding the commencement of recruitment to the study (International Monetary Fund, 2019). External data will also be extracted with regard to the 5 year average health care expenditure per capita PPP (current international \$) of the country in which each study takes place, incorporating the five years directly preceding the commencement of recruitment to the study (World Health Organisation, 2019). Each calendar year of the study will also be included provided recruitment continues through to > 6 months in the preceding year.”

Lines 222 - 226:

“Correlation analysis will also be employed to examine the relationship between the prevalence of frailty of geriatric inpatients and economic prosperity (GDP per capita PPP - current international \$), and health care expenditure (per capita PPP - current international \$). Additionally, multi-linear regression analysis will examine the predictive value between the economic prosperity and health care expenditure, and the prevalence of frailty amongst geriatric hospital inpatients.”

References:

International Monetary Fund, (. 2019, April-last update, World Economic Outlook (April 2019).

Available: <https://www.imf.org/external/datamapper/PPPPC@WEO/ADVEC/WEOWORLD> [2019, 02/05/].

World Health Organisation, (. 2019, April-last update, Global Health Expenditure Database (GHED).

Available: <http://apps.who.int/nha/database/Home/Index/en> [2019, 02/05/].

Comment - When will the study be done? How long time is it planned to take before it is completed?

Response - This study has been underway for several months. Reviewers have now completed the screening phase and are presently contacting 225 authors for additional information or clarification regarding inclusion as well as quality assessment of included studies. The planned completion date of this review was initially stated as June 2019 within the covering letter submitted alongside this manuscript but has now been revised to August 2019.

Comment - Page 6. Search strategy: for which years will the search be done?

Response - The search was last conducted on the 21/11/2018 and will encompass all years preceding this. However, in practice operation definitions for the classification of frailty are a relatively new phenomenon and only 10 of the 664 studies within the full text screening of this review were published prior to 2000, and >600 of the 664 studies were published since 2009. This has been added to the manuscript as follows on line 122 – 123 (additive text in bold):

“The search strategy will be conducted on the two platforms of Ovid and Web of Science, as well as the databases of SCOPUS, CINAHL Plus, and the Cochrane Library databases (Appendix 1). These searches will encompass all available literature published prior to 21/11/2018.”

Comment - Page 8. Data synthesis: “If a sufficient quantity of identified studies are comparable....”

How will you determine if this is accomplished? How many studies are needed, and how will you decide if they are comparable?

Response – In terms of comparability this is described in detail in the lines 193 – 196 (see below) and will be based on two reviewer’s opinion of the clinical heterogeneity between studies, with a third reviewer’s opinion being sought if a consensus cannot be reached between the two initial reviewers:

“Clinical heterogeneity will be assessed by two reviewers based on their judgement of the available data and any disagreements will be discussed thoroughly with the aim of reaching a unanimous consensus. If a unanimous consensus cannot be reached, the opinion of a third reviewer will be sought, and the proceeding majority consensus will be taken”.

Subsequent to this, provided there is agreement that there is a sufficient quantity of comparable studies, statistical heterogeneity will be assessed as described in lines 197 – 199 of the manuscript: “Statistical heterogeneity will be assessed through the utilisation of a Cochran Q test and considered present at $p < .05$. An I² test will be performed in order to assess the magnitude of this heterogeneity, with I² values of 25%, 50% and 75% being considered low, moderate and high respectively.”

As this review is concerned with pooled estimates of the prevalence of frailty, two or more is a sufficient quantity provided the two reviewers agree that the studies are comparable in terms of clinical heterogeneity to justify their collation as described above.

Comment - Page 9, line 181-182: why only two age-groups? I would prefer at least three, 65-74, 75-84 and 85+. The prevalence of frailty might/is very likely to be higher for 85+ than 75+. Which types of wards and morbidity will be included?

Response - The authors have considered this suggestion by the reviewer and agree to include more age groups, specifically those from 65 – 74, 75- 84 and 85+, this has now been amended in the protocol on line as follows (additive text in bold):

“Stratified analysis will also be conducted according to age (65 – 74 years, 75 – 84 years and 85+ years)”

All wards and all major shared morbidities within a cohort will be included to encompass not only a truly representative estimate of the overall prevalence of frailty in geriatric inpatients, but also one which provides a thoroughly detailed insight into this prevalence through stratified analysis.

Comment - The limitations of the study is stated, but not discussed (how will the limitations affect the results and the interpretation of the results of the study?).

Response - The stated limitation that this review will include only studies for which the full text is available in English will mean that otherwise potentially eligible studies where the full text is not available in English, will be omitted from this review. This is due to the practicality of all reviewers being native English speakers. Although a number have limited proficiency in other languages, to include any additional languages would require at a minimum all three independent reviewers during the screening phase to have working proficiency in a shared language other than English, which unfortunately is not the case. As such it is possible that studies published in non-English speaking countries which have not been translated into English will not be included. Although authors of these

studies will be contacted if for example a studies full text is not in English but can be identified as potentially eligible through an English title or abstract. Given this, it is possible that this present review will have a relative over-representation of studies from the Western world (Europe, Australasia and the Americas), as these places are more likely to publish findings in English, although they also represent the overwhelming majority of scientific publications in general.

The authors have briefly discussed this further on Line 52-54 (additive text in bold):

“Will include only studies for which the full text is available in English, therefore may be relatively over-representative of Western nations (Europe, Australasia, and the Americas); although this is true of scientific publications in general”

Reviewer: 3

Reviewer Name: Dr Rebecca L. Gould

Institution and Country: Senior Research Fellow, Division of Psychiatry, University College London, London, UK

Comment - This is a well written protocol for a systematic review in an important area. My comments are shown below:

Response - Thank you for your review of this manuscript. Please find a response to each of your comments below.

Strengths and limitations:

1) It says 3 independent reviewers will be involved, but 3 reviewers will only be involved in screening titles and abstracts. The most important tasks in a systematic review are data extraction and quality appraisal, and only 2 reviewers will be involved in these processes. So it is a misrepresentation to say that a strength of the study is “3 reviewers, ensuring high internal reliability and consistency”. This point needs to be removed as having 2 reviewers for the data extraction and quality appraisal tasks (with a third independent person to discuss any discrepancies that cannot be resolved through discussion with the 2 reviewers) is standard practice for systematic reviews (and the minimum requirement according to PRISMA guidelines). It is not a particular strength of this study. Instead, the authors could mention the comprehensive search of a wide range of literature sources being a strength of the study.

Response - The authors acknowledge the potential misinterpretation and have amended the manuscript as follows (additive text in bold):

“Three independent reviewers during the screening phase; ensuring high internal reliability and consistency of included studies”.

The authors would however like to highlight the importance of the screening phase in this review in terms of its rigour. The screening phase of this review was concluded recently, after several months of screening 664 full text article, culminating in over 19 hours of discussion of full text screening results between the three reviewers. Additionally, a majority of items relevant to quality assessment are included in the eligibility criteria, facilitating progression through screening of only high-quality studies.

Comment - 2) The last point is no different from the first point and therefore should be removed.

Response - We have now combined the first and last points on line 46 – 47 to read:

“First review to systematically or exclusively assess the overall prevalence of frailty in geriatric hospital inpatients

Comment - 3) The limitation of only including studies in English should be expanded upon – what will this mean for the interpretation of results and any conclusions drawn?

Response - The stated limitation that this review will include only studies for which the full text is available in English will mean that otherwise potentially eligible studies where the full text is not available in English, will be omitted from this review. This is due to the practicality of all reviewers

being native English speakers. Although a number have limited proficiency in other languages, to include any additional languages would require at a minimum all three independent reviewers during the screening phase to have working proficiency in a shared language other than English, which unfortunately is not the case. As such it is possible that studies published in non-English speaking countries which have not been translated into English will not be included. Although authors of these studies will be contacted if for example a studies full text is not in English but can be identified as potentially eligible through an English title or abstract. Given this, it is possible that this present review will have a relative over-representation of studies from the Western world (Europe, Australasia and the Americas), as these places are more likely to publish findings in English, although they also represent the overwhelming majority of scientific publications in general.

The authors have briefly discussed this further on Line 52-54 (additive text in bold):

“Will include only studies for which the full text is available in English, therefore will likely be relatively over-reprehensive of Western nations (Europe, Australasia, and the Americas); although this is true of scientific publications in general”

Comment - Eligibility criteria:

1) The authors note that all studies must have a minimum age of ≥ 65 years. Imagine that a study does not report a minimum age range, but instead reports a mean age of 80 years, will the study be excluded?

Response - Only studies which can confirm the minimum age of participants will be included. The authors of any studies within the review where a minimum age is not expressly stated, or there is some degree of ambiguity in this regard, will be contacted for clarification of the minimum age of participants. If the minimum age cannot be obtained, then the study will be excluded irrespective of mean age.

Comment - Studies frequently report mean ages rather than age range, and they may not report age as an inclusion criterion if they've examined geriatric inpatient wards (e.g. they might just include anybody on the geriatric inpatient wards, and wards may vary in terms of how they define “geriatric” e.g. some may say 60 years and over). The authors may want to consider specifying a minimum mean age, as well as the minimum age, or at the very least acknowledge the limitation of this approach.

Response - The authors are reluctant to include a minimum mean age as this approach in many cases is likely to include those who are not within the target population for this review i.e. geriatric inpatients (individuals >64 years), and as such would negatively impact upon the integrity of the results of the review. As mentioned above the authors of such studies will be contacted to confirm the minimum age of participants where this is not expressly stated.

Comment - 2) How are the authors defining “a clearly defined and validated operational definition for the classification of frailty”? Do they mean use of a validated frailty assessment tool, a self-rating questionnaire, etc? This needs to be discussed in more detail.

Response - This relates to any operational definition of frailty which has been specifically validated for the assessment of frailty, either through comparison with existing validated frailty tools or its predictive value regarding negative health outcomes aligned with frailty. The reason for this particular eligibility criterion is that within the literature there are a plethora of studies which either subjectively describe patients as frail using no clearly defined criteria, or operationally define patients as frail utilising clearly defined criteria which have not been validated specifically with respect to their validity in the operational definition of frailty. We have added text regarding this on line 92-94:

“(i.e. has been specifically validated for the assessment of frailty, either through comparison with existing validated frailty tools, or its predictive value regarding negative health outcomes aligned with frailty”.

Comment - 3) The authors give outpatients, day patients or community-dwelling individuals as examples of non-hospital inpatients. Presumably they mean non-hospital outpatients?

Response - The authors believe the reviewer has misread this line. However, this may be due to a degree of ambiguity in the phrasing. We have now clarified this as follows with the removal of the term

“non-hospital inpatient” and replaced this with “are not hospital inpatients” on Line 100 (altered text in bold):

“Exclusion criteria: all studies not written in English, studies where the sample are not hospital inpatients (i.e. outpatients, day patients or community-dwelling individuals).”

Comment - Information sources:

1) The authors need to provide the years that will be searched for each of the databases.

Response - The search has last been conducted on the 21/11/2018 and will encompass all years proceeding this. However, in practice operation definitions for the classification of frailty are a relatively new phenomenon and only 10 of the 664 studies within the full text screening of this review were published prior to 2000, and >600 of the 664 studies were published since 2009. This has been added to the manuscript as follows on line 122 – 124 (additive text in bold):

“The search strategy will be conducted on the two platforms of Ovid and Web of Science, as well as the databases of SCOPUS, CINAHL Plus, and the Cochrane Library databases (Appendix 1). These searches will encompass all available literature published prior to 21/11/2018.”

Comment - 2) There is no mention of manual searching (e.g. hand searching of references of relevant studies and literature reviews). Why are the authors not doing this? This is standard practice as some older studies may not have been entered onto the databases.

Response - Manual searching is being employed by authors. All reference lists of included studies are to be screened. Additionally, reference list of excluded, but potentially interesting studies or systematic reviews captured within the screening, are also to be searched. As part of the grey literature search, in process publication have also been searched and conference abstracts will be followed up with the authors to see if a sufficient degree of information exists to include the studies in the review. This has now been clarified within the manuscript on line 147 - 150:

“Manual screening will also be employed by reviewers and include the reference lists of all included studies, as well as excluded but potentially relevant studies or systematic reviews captured within the screening. As part of the grey literature search of this review, in process publications have also been searched and conference abstracts will be followed up with authors to ascertain if a full text relating to the data is available.”

Comment - Data extraction:

1) Why are the authors not extracting data regarding ethnicity? Some studies may not provide these data, but this doesn't mean that you should not attempt to extract these data.

Response - Authors are not extracting data regarding ethnicity as it is not present within the overwhelming majority of these studies. Although a potentially interesting area of inquiry, and one which the authors did initially consider examining as it has been shown to be a variable associated with differences in the prevalence of frailty, through examination of preliminary studies identified prior to the implementation of systematic searches, it was apparent that such data are generally absent within these studies. This remains the case post full text screening for vast majority of studies. In this regard the authors are cognisant of some of the pitfalls that would arise and impinge upon the integrity of the results of the review if attempting to continue in this regard with a profoundly absent dataset, particularly as this review is concerned with the worldwide prevalence of frailty. An example of one such pitfall, which would be ubiquitous in proceeding in this manner, would be a situation where one study of 400 inpatients in Japan (a relatively homogenous country), does not report ethnicity, and another study in the UK which does report ethnicity and comprised of 50 ethnically Japanese individuals. This would result in this review analysing data from the latter study in the UK and claiming it as representative of the prevalence of frailty amongst people of Japanese ethnicity, while the former is excluded.

Comment - 2) Presumably the authors will extract data regarding the criteria utilised for the operational definition of pre-frailty, as well as frailty, given that they are also interested in the pre-frail state?

Response - The specific tool itself will be extracted as opposed to the specific criteria which constitutes the tool within the data extraction form. The tool itself utilised in this regard will be the same for assessment of frailty/pre-frailty. However, not all frailty screening tools classify in this manner (frail/pre-frail/robust (or non-frail)), as such this is likely be a reduced proportion of studies.

Comment - Data synthesis:

1) Why did the authors choose age, sex, frailty definition, ward type, prevalent morbidity and location as factors for the stratified analysis? Are these evidence-based factors that have been shown to influence prevalence rates in other frailty populations? There needs to be more justification for this choice of variables, either in the Introduction or Methods section.

Response - These factors were chosen predominantly due to:

1. an enhanced knowledge of these specific areas being of practical utility to researchers and clinicians
2. empirical evidence showing alterations in these variables have an impact upon the prevalence of frailty, as such allowing this review to provide a more in-depth and thorough insight into the prevalence of frailty stratified by these variables.
3. the ubiquitous availability of data pertaining to these variables within studies in this area.

A justification of this has been added as follows on lines 204 - 210 (additive text in bold):

“Stratified analysis will also be conducted according to age (65 – 74 years, 75 – 84 years and 85+ years), sex, frailty definition, ward type, prevalent morbidity and location where possible. These variables have been specifically chosen for stratified analysis predominantly due to an enhanced knowledge of these areas being of particular utility to researchers and clinicians; stemming from empirical evidence persistently showing alterations in these variables to impact upon prevalence of frailty (Fried et al. 2001, Purser et al. 2006, Santos-Eggimann et al. 2009, Andela et al. 2010, Sternberg et al. 2011). As such stratified analysis pertaining to these variables will facilitate this review in providing a more in-depth and thorough insight into the prevalence of frailty amongst geriatric hospital inpatients.”

References:

- Fried, L.P., Tangen, C.M., Walston, J., Newman, A.B., Hirsch, C., Gottdiener, J., Seeman, T., Tracy, R., Kop, W.J. & Burke, G. 2001, "Frailty in older adults evidence for a phenotype", *The Journals of Gerontology Series A: Biological Sciences and Medical Sciences*, vol. 56, no. 3, pp. M157.
- Purser JL, Kuchibhatla MN, Fillenbaum GG, Harding T, Peterson ED, Alexander KP. Identifying frailty in hospitalized older adults with significant coronary artery disease. *J Am Geriatr Soc* 2006 NOV;54(11):1674-1681.
- Santos-Eggimann B, Cuénoud P, Spagnoli J, Junod J. Prevalence of frailty in middle-aged and older community-dwelling Europeans living in 10 countries. *The Journals of Gerontology: Series A* 2009;64(6):675-681.
- Andela RM, Dijkstra A, Slaets JPJ, Sanderman R. Prevalence of frailty on clinical wards: Description and implications. *Int J Nurs Pract* 2010 FEB;16(1):14-19.
- Sternberg, S.A., Schwartz, A.W., Karunanathan, S., Bergman, H. & Mark Clarfield, A. 2011, "The identification of frailty: a systematic literature review", *Journal of the American Geriatrics Society*, vol. 59, no. 11, pp. 2129-2138.

Comment - 2) More information needs to be provided about how some of the variables will be stratified. Will frailty definition be stratified by frailty model (e.g. cumulative deficits model vs. Fried model) or assessment mode (e.g. self-rating vs. clinician-rating of frailty)?

Response - As outlined within the manuscript on lines 36, 48-49, 80, 87, and 205 the prevalence of frailty will be stratified by “frailty definition”, referring specifically to the operational definition of frailty

utilised i.e. the validated tool for the assessment of frailty utilised. We have added the word “operational” to prefix all of these above referenced instances to help clarify this.

Comment - How will prevalent morbidity be stratified?

Response - Prevalent morbidity / clinical population will be stratified by shared prevalent morbidity. For example, if there are several studies including patients undergoing cardiac valve replacement, provided two reviewers have agreed on the clinical heterogeneity of these studies, stratified analysis will be performed on this group of studies based on the shared prevalent morbidity.

Comment - Does location mean country? How will this be stratified?

Response - Location specifically refers to the country and continent on which the study takes place. As such the prevalence of frailty will be stratified by country and continent, in a manner consisted with the data extraction form attached to this application (Appendix 5 – column N and O respectively). The authors have now also included a component within the review, assessing the association between the economic prosperity, and health care expenditure of countries, and the prevalence of frailty of geriatric hospital inpatients, which has been included from lines 183 – 189 and lines 222 – 226 of the manuscript (additive text outlined below) and within the data extraction form in Appendix 5 (Columns P and Q of spreadsheet one, and spreadsheet 2 and 3 in their entirety).

Lines 183 - 189:

“External to the studies, data will also be extracted with regard to the 5 year average Gross Domestic Product (GDP) per capita Purchasing Power Parity (PPP) (current international \$) of the country in which each study takes place; incorporating the five year directly preceding the commencement of recruitment to the study (International Monetary Fund, 2019). External data will also be extracted with regard to the 5 year average health care expenditure per capita PPP (current international \$) of the country in which each study takes place, incorporating the five years directly preceding the commencement of recruitment to the study (World Health Organisation, 2019). Each calendar year of the study will also be included provided recruitment continues through to > 6 months in the preceding year.”

Lines 222 - 226:

“Correlation analysis will also be employed to examine the relationship between the prevalence of frailty of geriatric inpatients and economic prosperity (GDP per capita PPP - current international \$), and health care expenditure (per capita PPP - current international \$). Additionally, multi-linear regression analysis will examine the predictive value between the economic prosperity and health care expenditure, and the prevalence of frailty amongst geriatric hospital inpatients.”

References:

International Monetary Fund, (. 2019, April-last update, World Economic Outlook (April 2019).

Available: <https://www.imf.org/external/datamapper/PPPPC@WEO/ADVEC/WEOWORLD> [2019, 02/05/].

World Health Organisation, (. 2019, April-last update, Global Health Expenditure Database (GHED).

Available: <http://apps.who.int/nha/database/Home/Index/en> [2019, 02/05/].

Comment - 3) More information needs to be provided about how they will complete the stratified analysis (e.g. subgroup analysis, metaregression, etc).

Response - Stratified analysis will follow essentially the same procedures as the overall pooled analysis. The following has now been added to lines 211 - 220 (additive text in bold):

“Clinical heterogeneity for stratified analysis will be assessed by two reviewers based on their judgement of the available data. Any disagreements will be discussed thoroughly with the aim of reaching a unanimous consensus. If a unanimous consensus cannot be reached, the opinion of a third reviewer will be sought. Statistical heterogeneity for sub-analysis will similarly be assessed through the utilisation of a Cochran Q test and considered present at $p < .05$. An I² test will be performed in order to assess the magnitude of this heterogeneity, with I² values of 25%, 50% and 75% being considered low, moderate and high respectively.

Similarly, it is likely a random-effects model will be utilised to synthesise pooled estimates of the prevalence of frailty stratified by these criteria (although there is more of a likelihood that a fixed

effects model could potentially be utilised within these analyses, in comparison to the initial analysis, given the nature of stratified analysis).”

Thank you for your comments. The authors hope the above adequately address these comments and thank you again for your review.

VERSION 2 – REVIEW

REVIEWER	Han Ting Wang Hopital Maisonneuve-rosemont
REVIEW RETURNED	22-May-2019

GENERAL COMMENTS	Again, thank you for the invitation to review this protocol. it was an already well written one and i think it is now much better. all my questions and comments have been addressed and i have no further inquiries. thank you!
---

REVIEWER	Katarina Wilhelmson Centre of Aging and Health, AgeCap, Department of Health and Rehabilitation, Institute of Neuroscience and Physiology, Sahlgrenska Academy, University of Gothenburg, Gothenburg, Sweden
REVIEW RETURNED	12-Jun-2019

GENERAL COMMENTS	Still not clear what is meant by “location”? Type of ward, type of hospital, city, country...? Still unclear when the study will be performed. To me, still unclear how you will determine that you have sufficient quantity of identified studies that are comparable. I still lack a discussion how the limitations of the study affects the results and the interpretation of the results. Information have been added concerning “Correlation analysis will also be employed to examine the relationship between the prevalence of frailty of geriatric inpatients and economic prosperity (GDP per capita PPP) (current international \$), and health care expenditure (per capita PPP) (current international \$). Additionally, multi-linear regression analysis will examine the predictive value between economic prosperity and health care expenditure, and the prevalence of frailty of geriatric inpatients.” What is the rational for these analyses?
---

REVIEWER	Rebecca Gould University College London, UK
REVIEW RETURNED	05-Jun-2019

GENERAL COMMENTS	The authors have responded well to the majority of my comments. I just have one comment arising from their responses. The authors note that they are not extracting data regarding ethnicity as it is not present within the overwhelming majority of the studies. I think it is still important to collect these data in order to highlight the weaknesses with respect to data collection in this area to the reader. If data collection is sparse then you, of course, would not infer anything from these data (the authors gave an example of
--

	claiming data from a study in the UK of Japanese individuals with ethnicity data being representative of the prevalence of frailty in this population). A conclusion that "no conclusions can be drawn with respect to ethnicity due to sparse data collection" (and therefore suggestive of an area of concern for future research) is still a valid conclusion, but something that cannot be drawn if you don't collect the data (and have the evidence to show this).
--	--

VERSION 2 – AUTHOR RESPONSE

Reviewer: 1

Reviewer Name: Han Ting Wang

Institution and Country: Hopital Maisonneuve-rosemont

Please state any competing interests or state 'None declared': none

Please leave your comments for the authors below

Comment - Again, thank you for the invitation to review this protocol.

it was an already well written one and i think it is now much better.

all my questions and comments have been addressed and i have no further inquiries.

thank you!

Response – Dear Dr. Wang, Thank you again for your review.

Reviewer: 3

Reviewer Name: Rebecca Gould

Institution and Country: University College London, UK

Please state any competing interests or state 'None declared': None

Please leave your comments for the authors below

Comment - The authors have responded well to the majority of my comments. I just have one comment arising from their responses. The authors note that they are not extracting data regarding ethnicity as it is not present within the overwhelming majority of the studies. I think it is still important to collect these data in order to highlight the weaknesses with respect to data collection in this area to the reader. If data collection is sparse then you, of course, would not infer anything from these data (the authors gave an example of claiming data from a study in the UK of Japanese individuals with ethnicity data being representative of the prevalence of frailty in this population). A conclusion that "no conclusions can be drawn with respect to ethnicity due to sparse data collection" (and therefore

suggestive of an area of concern for future research) is still a valid conclusion, but something that cannot be drawn if you don't collect the data (and have the evidence to show this).

Response - Dear Dr. Gould, Thank you again for your review. After a careful consideration of the above point, which the authors believe is very well-made, we would like to remain consistent in our rationale for stratified analysis, that is to say that, the stratified variables were due to:

1. an enhanced knowledge of these specific areas being of practical utility to researchers and clinicians
2. empirical evidence showing alterations in these variables have an impact upon the prevalence of frailty, as such allowing this review to provide a more in-depth and thorough insight into the prevalence of frailty stratified by these variables.
3. the ubiquitous availability of data pertaining to these variables within studies in this area.

While the first two of these conditions are met with regard to ethnicity, the third is not. While we appreciate it is important to note areas which require further investigation, given that we know these data are not available, it is possible to facilitate this with a narrative commentary to this effect within the results paper when discussing areas for further investigation; as this meta-analysis is primarily concerned with examining the data which is available regarding the prevalence of frailty amongst geriatric hospital inpatients. However, we agree with the Reviewer's point, and within the discussion section of the results paper there is room to mention why certain variables weren't examined, for example ethnicity, and call for more research to be conducted in this regard to facilitate this.

Reviewer: 2

Reviewer Name: Katarina Wilhelmson

Institution and Country:

Centre of Aging and Health, AgeCap, Department of Health and Rehabilitation, Institute of Neuroscience and Physiology, Sahlgrenska Academy, University of Gothenburg, Gothenburg, Sweden

Please state any competing interests or state 'None declared': None declared

Please leave your comments for the authors below

Comment - Still not clear what is meant by "location"? Type of ward, type of hospital, city, country...?

Response – Dear Dr. Wilhelmson, thank you again for your review. The authors feel this has been addressed in the initial response:

"Location specifically refers to the country and continent on which the study takes place. As such the prevalence of frailty will be stratified by country and continent, in a manner consistent with the data extraction form attached to this application (Appendix 5 – column N and O respectively). We have also now changed "country/location" to "country/continent" on line 173."

We have now also clarified this within the main text each time the term "location" is mentioned, as follows on lines 80 and 204 (additive text in bold):

“location (country and continent)”

Comment - Still unclear when the study will be performed.

Response – The authors feel this has been addressed in the initial response:

“This study has been underway for several months. Reviewers have now completed the screening phase and are presently contacting 225 authors for additional information or clarification regarding inclusion as well as quality assessment of included studies. The planned completion date of this review was initially stated as June 2019 within the covering letter submitted alongside this manuscript but has now been revised to August 2019.”

In addition to the insertion of the following line in the initial response on lines 122-123: “These searches will encompass all available literature published prior to 21/11/2018” we have now also added the following on lines 243:

“This review is scheduled for completion during the second half of 2019”

Comment - To me, still unclear how you will determine that you have sufficient quantity of identified studies that are comparable.

Response – The initial response was “– In terms of comparability this is described in detail in the lines 193 – 196 (see below) and will be based on two reviewers opinion of the clinical heterogeneity between studies, with a third reviewer’s opinion being sought if a consensus cannot be reached between the two initial reviewers:

“Clinical heterogeneity will be assessed by two reviewers based on their judgement of the available data and any disagreements will be discussed thoroughly with the aim of reaching a unanimous consensus. If a unanimous consensus cannot be reached, the opinion of a third reviewer will be sought, and the proceeding majority consensus will be taken”.

Subsequent to this, provided there is agreement that there is a sufficient quantity of comparable studies, statistical heterogeneity will be assessed as described in lines 197 – 199 of the manuscript:

“Statistical heterogeneity will be assessed through the utilisation of a Cochran Q test and considered present at $p < .05$. An I² test will be performed in order to assess the magnitude of this heterogeneity, with I² values of 25%, 50% and 75% being considered low, moderate and high respectively.”

As this review is concerned with pooled estimates of the prevalence of frailty, two or more is a sufficient quantity provided the two reviewers agree that the studies are comparable in terms of clinical heterogeneity to justify their collation as described above.”

As such this determination is based on the two independent data extractors informed subjective opinions regarding the heterogeneity of two or more studies. Several factors will be taken into account when determining this heterogeneity, including the demographic of the samples and the setting in which the study takes place. Given the inclusion criteria of this review, comparability with regard to the overall pooled estimate is essentially assured in this regard for all included studies, although it is likely to utilise a random-effects model.

Comment - still lack a discussion how the limitations of the study affects the results and the interpretation of the results.

Response - The authors feel this has been addressed in the initial response:

“The stated limitation that this review will include only studies for which the full text is available in English will mean that otherwise potentially eligible studies where the full text is not available in English, will be omitted from this review. This is due to the practicality of all reviewers being native English speakers. Although a number have limited proficiency in other languages, to include any additional languages would require at a minimum all three independent reviewers during the screening phase to have working proficiency in a shared language other than English, which unfortunately is not the case. As such it is possible that studies published in non-English speaking countries which have not been translated into English will not be included. Although authors of these studies will be contacted if for example a studies full text is not in English but can be identified as potentially eligible through an English title or abstract. Given this, it is possible that this present review will have a relative over-representation of studies from the Western world (Europe, Australasia and the Americas), as these places are more likely to publish findings in English, although they also represent the overwhelming majority of scientific publications in general.

The authors have briefly discussed this further on Line 52-54 (additive text in bold):

“Will include only studies for which the full text is available in English, therefore may be relatively over-representative of Western nations (Europe, Australasia, and the Americas); although this is true of scientific publications in general”

Comment - Information have been added concerning “Correlation analysis will also be employed to examine the relationship between the prevalence of frailty of geriatric inpatients and economic prosperity (GDP per capita PPP) (current international \$), and health care expenditure (per capita PPP) (current international \$). Additionally, multi-linear regression analysis will examine the predictive value between economic prosperity and health care expenditure, and the prevalence of frailty of geriatric inpatients.” What is the rational for these analyses?

Response – Preliminary research into these areas have shown frailty in the community to be correlated with economic indicators (Gross Domestic Product per capita Price Purchasing Parity (GDP PPP)) (1), however, note that more research is needed in this regard to better understand this relationship, which this review will facilitate in regard to the relationship of GDP PPP, and health care expenditure and the prevalence of frailty amongst geriatric hospital inpatients.

References:

(1) Theou O, Brothers TD, Rockwood MR, Haardt D, Mitnitski A, Rockwood K. Exploring the relationship between national economic indicators and relative fitness and frailty in middle-aged and older Europeans. *Age Ageing* 2013;42(5):614-619.

We thank the editor, and once again all the reviewers, for their time in the review of this manuscript and hope that all of the above comments have been adequately addressed, and that the manuscript is now suitable for publication.

Yours Sincerely,

Mr. Paul Doody, Mr. Justin Aunger, Mr. Evans Asamane, Dr. Carolyn Greig, Professor Janet Lord, Professor Anna Whittaker.

VERSION 3 – REVIEW

REVIEWER	Katarina Wilhelmson Centre of Aging and Health-AgeCap, Department of Health and Rehabilitation, Institute of Neuroscience and Physiology, Sahlgrenska Academy, University of Gothenburg, Gothenburg, Sweden
REVIEW RETURNED	04-Jul-2019

GENERAL COMMENTS	Thank you for your additional revision and clarifying response! Concerning how you will determine that you have sufficient quantity of identified studies that are comparable, I could have been clearer in my comment. It was how you determine the clinical heterogeneity between the studies (“based on their judgment of the available data”) that I lacked information about. Do you use some kind of checklist, or how will you determine the clinical heterogeneity? The assessment of the statistical heterogeneity is well described. Concerning the added text about the economic indicators, you give a good rationale and reference for this in your response, but I cannot find this stated in the manuscript? Why not include this information in the manuscript?
---

VERSION 3 – AUTHOR RESPONSE

Reviewer(s)' Comments to Author:

Reviewer: 2

Reviewer Name: Katarina Wilhelmson

Institution and Country: Centre of Aging and Health-AgeCap, Department of Health and Rehabilitation, Institute of Neuroscience and Physiology, Sahlgrenska Academy, University of Gothenburg, Gothenburg, Sweden

Please state any competing interests or state ‘None declared’: None declared

Please leave your comments for the authors below

Comment - Thank you for your additional revision and clarifying response! Concerning how you will determine that you have sufficient quantity of identified studies that are comparable, I could have been clearer in my comment. It was how you determine the clinical heterogeneity between the studies (“based on their judgment of the available data”) that I lacked information about. Do you use some kind of checklist, or how will you determine the clinical heterogeneity? The assessment of the statistical heterogeneity is well described.

Response – Dear Dr. Wilhelmson, Thank you again for you review and clarification with regard to your initial comments. As mentioned in the previous response “this determination is based on the two independent data extractor’s informed subjective opinions regarding the heterogeneity of two or more studies. Several factors will be taken into account when determining this heterogeneity, including the

demographic of the samples, and the settings in which the studies take place. Given the inclusion criteria of this review, comparability with regard to the overall pooled estimate is essentially assured in this regard for all included studies”.

The above aforementioned procedure relating to the assessment of clinical heterogeneity within this review is consistent with Preferred Reporting Items for Systematic Review and meta-analysis Protocols (PRISMA-P) 2015; which provide the following example “We will test the clinical heterogeneity by considering the variability in participant factors among trials (for example age) and trial factors” (1). The authors are not aware of the existence of a specific clinical heterogeneity checklist relevant to this review (though are aware of previous calls for the creation of such a checklist (2).

We hope that this has sufficiently clarified this aspect of the manuscript for the reviewer: specifically that the assessment of clinical heterogeneity is based on the two independent data extractors informed subjective opinions regarding the heterogeneity of two or more studies, and that several factors will be taken into account when determining this heterogeneity, including the demographic of the samples and the settings in which the studies take place; in a manner consistent with PRISMA-P guidelines (1).

References

(1) Shamseer L, Moher D, Clarke M, Ghersi D, Liberati A, Petticrew M, et al. Preferred reporting items for systematic review and meta-analysis protocols (PRISMA-P) 2015: elaboration and explanation. *BMJ* 2015;349:g7647.

(2) Gagnier JJ, Moher D, Boon H, Beyene J, Bombardier C. Investigating clinical heterogeneity in systematic reviews: a methodologic review of guidance in the literature. *BMC medical research methodology* 2012;12(1):111.

Comment - Concerning the added text about the economic indicators, you give a good rationale and reference for this in your response, but I cannot find this stated in the manuscript? Why not include this information in the manuscript?

Response – We have now added this to the manuscript on lines 225 – 229 as follows:

“Preliminary research into these areas have shown frailty in the community to be correlated with economic indicators (GDP per capita PPP) (23), however, note that more research is needed in this regard to better understand this relationship; which this review will facilitate through examination of the relationship between GDP per capita PPP and health care expenditure, and the prevalence of frailty amongst geriatric hospital inpatients.”

References:

(3) Theou O, Brothers TD, Rockwood MR, Haardt D, Mitnitski A, Rockwood K. Exploring the relationship between national economic indicators and relative fitness and frailty in middle-aged and older Europeans. *Age Ageing* 2013;42(5):614-619.

We once again thank the editor and all the reviewers for their time in the review of this manuscript and hope that their comments have been adequately addressed, and that the manuscript is now suitable for publication.